# The Role of p53-Mediated Signaling in the Therapeutic Response of Colorectal Cancer to 9F, a Spermine-Modified Naphthalene Diimide Derivative

**DOI:** 10.3390/cancers12030528

**Published:** 2020-02-25

**Authors:** Lei Gao, Chaochao Ge, Senzhen Wang, Xiaojuan Xu, Yongli Feng, Xinna Li, Chaojie Wang, Yuxia Wang, Fujun Dai, Songqiang Xie

**Affiliations:** 1Key Laboratory of Natural Medicine and Immuno-Engineering, Henan University, Kaifeng 475004, Henan, China; gaolei@vip.henu.edu.cn (L.G.); gechao1234365@163.com (C.G.); wangsz8836@163.com (S.W.); fyl062063@163.com (Y.F.); 17719096749@163.com (X.L.); wcjsxq@henu.edu.cn (C.W.); 2Pharmaceutical College, Henan University, Kaifeng 475004, Henan, China; xuxiaojuahn@163.com; 3College of Chemistry and Chemical Engineering, Henan University, Kaifeng 475004, Henan, China; wangyuxia@henu.edu.cn

**Keywords:** naphthalimide, polyamine, colorectal cancer, p53

## Abstract

Colorectal cancer (CRC) is one of the most prevalent cancers due to its frequency and high rate of mortality. Polyamine-vectorized anticancer drugs possess multiple biological properties. Of these drugs, 9F has been shown to inhibit tumor growth and the metastasis of hepatocellular carcinoma. This current study aims to investigate the effects of 9F on CRC and determine its molecular mechanisms of action. Our findings demonstrate that 9F inhibits CRC cell growth by inducing apoptosis and cell cycle arrest, and suppresses migration, invasion and angiogenesis in vitro, resulting in the inhibition of tumor growth and metastasis in vivo. Based on RNA-seq data, further bioinformatic analyses suggest that 9F exerts its anticancer activities through p53 signaling, which is responsible for the altered expression of key regulators of the cell cycle, apoptosis, the epithelial-to-mesenchymal transition (EMT), and angiogenesis. In addition, 9F is more effective than amonafide against CRC. These results show that 9F can be considered as a potential strategy for CRC treatment.

## 1. Introduction

Colorectal cancer (CRC) is one of the mostmalignant types of cancers that have been increasing in prevalence on a global scale. Due to its high rate ofmortality, CRC is one of the main causes of cancer deathin the world [1]. Patients with CRC are usually diagnosed with distant metastasis, and the 5-year survival rate is no more than 10% in patients with advanced-stage CRC [2].Over the years, variousstrategiesincluding chemotherapy (i.e., cytotoxic and molecular targeted agents) have been applied to improve the treatment results of patients with CRC [3,4].Despite advances in early detection and treatment, the number of patients with CRC will still grow in the coming decades [5].Thus, exploringnovel therapies is crucial for improving clinical outcomes in patients with CRC.

As versatile functional compounds, naphthalimides possess many therapeutic properties such as anticancer and antiviral activities [6,7,8]. Several naphthalimide derivatives have been shown to exert many inhibitory effects on cancer cells through various mechanisms [9,10,11,12]. Notably, several naphthalimides, such as amonafide, mitonafide, and elinafide [13,14,15], have been evaluated clinically as promising anti-cancer agents. For example, amonafide was the first naphthalimide evaluated in clinical trials alone [16,17] and in combination with other anti-cancer drugs [18]. In recent years, agents composed of a polyamine skeleton have displayed various biological properties [19,20,21,22]. Supporting the potential importance of the polyamine scaffold, anew spermine-vectorized anticancer drug (F14512) has been evaluated as an anticancer candidate in phase I clinical trials [23,24,25].

As a classical tumor suppressor, p53 has been investigated extensively and has been known to play a critical role in tumor growth and metastasis [26]. In response to stress, p53 acts as a transcriptional factor that regulates the expression of downstream target genes involved in the cell cycle, apoptosis, DNA damage and senescence [27,28]. A correlation has been identified between p53 and the death receptor [29,30], which suggests a significant role of p53 in death receptor-mediated cell death. Metastasis is the main cause of cancer death, and the epithelial-to-mesenchymal transition (EMT) is a crucial process for metastasis [31,32]. Recently, emerging evidence has demonstrated that p53 acts as a negative regulator of EMT via by mediating the expression of EMT markers [33,34] such as Snail, Twist, Slug and Vimentin [35,36,37,38]. These results imply that p53 loss or mutation creates a permissive environment for metastasis [39].

In order to increase the antitumor efficacy and reduce the adverse effects of naphthalimide derivates, we developed several naphthalimide-polyamine conjugates and found evidence of their anticancer activities in experimental models using several cancer types [40,41,42]. In the present study, we report that 9F, a naphthalene diimide derivative with polyamine moieties, inhibits the growth and lung metastasis of CRC in vitro and in vivo. Using RNA-seq, 9F was identified to induce cell cycle arrest, apoptosis, and oppose EMT as well as angiogenesis by activating p53-mediated signaling.

## 2. Results

### 2.1. 9F Inhibits Cell Growth and Induces Cell Deathof CRC

The chemical structure of 9F is shown in Figure 1A. To understand the effects of 9F on CRC, we examined the inhibitory effects of 9F on several cell lines including the CT-26, HCT116 and HCT8 cells. As displayed in Figure 1B, the cell viability of these three cell lines was obviously decreased after treatment with 9F for 48 h. Meanwhile, 9F inhibited the growth rate of CRC cells to a greater extent than amonafide, which was used as a positive control (Figure 1B). To confirm the anticancer properties of 9F, CRC cell lines (i.e., CT-26 and HCT116) were exposed to 9F at various concentrations for 24 h, 48 h and 72 h. As shown in Figure 1C, treatment with 9F caused a significant inhibition of cell viability in a time- and concentration-dependent manner. To investigate whether the inhibitory effects of 9F on the growth of CRC cells were sustained beyond three days, the anti-proliferative effects of 9F were detected using 2D colony formation assays. Treatment of CRC cells with 9Fremarkably reduced the number of cell colonies (Figure 1D). Consistently, both dual acridine orange/ethidium bromide (AO/EB) staining and dual Annexin V/propidium iodide (PI) staining revealed an increased rate of cell death in 9F-treated groups, compared to the control groups (Figure 1E,F). Taken together, these results demonstrated the inhibition of cell growth and induction of cell death by 9F in CRC cells.

### 2.2. Identification of p53-Mediated Pathway in 9F-Treated CRC Cells

To decipher the potential mechanisms underlying the inhibitory effects of 9F on CRC cells, RNA-seq was carried out to profile changes in the transcriptome of HCT116 cells after treatment with 9F. Based on the results of MTT assays, cells were treated with 9F (1 μM) for 24 h. To analyze the differentially expressed genes (DEGs), the following criteria were applied: log2 fold changes (FCs) > 1 and a false detection rate (FDR) of less than 0.05. In total, we identified 1217 genes with significantly altered expression in HCT116 cells treated by 9F. Among these DEGs, 564 genes were upregulated, and 653 genes were downregulated (Figure 2A). Most importantly, the Kyoto Encyclopedia of Genes and Genomes (KEGG) pathway enrichment analysis for the DEGs displayed thetop 20 pathways that were enriched. Among these pathways, signaling cascades affecting the cell cycle, DNA replication, the p53 signaling pathway, and apoptosis were significantly enriched (Figure 2B). Thep53 signaling pathway plays a critical role in the regulation of the cell cycle, DNA replication and apoptosis [27,28]. Therefore, we speculated that the p53 signaling pathway is the main determinant responsible for the inhibitory effects of 9F on CRC. We performed gene set enrichment analysis (GSEA), a powerful analytical method for interpreting genome-wide expression profiles, to test this hypothesis and to obtain an unbiased result. As shown in Figure 2C, the GSEA results implied that the significant hallmark gene set that was enriched in 9F-treated cells was implicated in p53 signaling. In addition, the p53 downstream pathway was also enrichedin 9F-treated cells. As a tumor suppressor, p53 exerts its functions by mediating the transcription of a series of genes. Considering this aspect, DEGs were analyzed using the Reactome database. In 9F-treated cells, the genes regulated by p53 are involved in cell death, the cell cycle, cellular metabolism and DNA repair (Figure 2C). The nuclear accumulation of p53 is required for the transcription of genes, and we found that nuclear localization of p53 increased in 9F-treated CRC cells (Figure 2D). Western blot results indicated that 9F increased the protein expression of p53, p21, and sestrin 2 (SESN2) in p53-wildtype HCT116 cells (Figure 2E). In contrast, 9F failed to activate the expression of p21 and SESN2 in p53-null HCT116 cells, thus suggesting that 9F induces the expression of p21 and SESN2 by a p53-dependent pathway. These results indicated that p53 and its target genes may contribute to the inhibitory effects of 9F on CRC cells.

### 2.3. 9F Induces Cell Cycle Arrest and Disrupts DNA Replication of CRCCells

The Reactome database was used to analyze DEGs. As shown in Figure 3A, the pathways participating in the cell cycle, DNA replication and DNA repair were obviously altered in 9F-treated HCT116 cells. Next, DEGs were further subjected to GO enrichment analysis, which identified the list of genes that were enriched in the cell cycle, the cell cycle process, the mitotic cell cycle process, mitotic cell cycle, cell division, DNA replication, sister chromatid segregation, the DNA metabolic process, nuclear division, and the chromosome segregation process (Figure 3B). These biological processes are involved in the cell cycle and DNA replication in HCT116 cells following exposure to 9F. In line with the above findings, using GSEA, there was a significant reduction in expression of genes associated with the cell cycle, G2/M checkpoint, DNA replication, and the E2F pathway in HCT 116 cells after treatment with 9F (Figure 3C). Among these dysregulated genes, CDKN1A, MDM2, GADD45A, GADD45B, CDKN2B and TMEM40 showed increased expression after 9F treatment. In contrast, CDC25A, CDC25C, CDC45, CDC6, CCNB1, CCNB2, CCNA2, CCNE2, E2F1, E2F2, WEE1, and minichromosome maintenance (MCM) proteins -2 to -7 were rapidly downregulated in HCT116 cells after 9F treatment (Figure 3D). Having shown that 9F mediates the expression of genes related to the cell cycle based on the data from RNA-seq, we questioned whether 9F induces cell cycle arrest. As expected, flow cytometry showed that treatment with 9F resulted in G2/M arrest among HCT116 cells and CT-26 cells (Figure 3E). These results were consistent with the GSEA results showing that 9F downregulated the expression of genes involved in the G2/M checkpoint (Figure 3C). Interestingly, as demonstrated by the western blot assay, the protein levels of GADD45A were remarkably increased, while MCM4, MCM7, Wee1, Cdc25C, Cyclin A, Cyclin B, Cyclin E, and proliferating-cell nuclear antigen (PCNA) protein levels were evidently reduced after 9F treatment in HCT116 cells (Figure 3F), thus confirming the data of the RNA-seq analysis. These results indicated that cell cycle arrest and disruption of DNA replication were induced by 9F in CRC cells.

### 2.4. Both the Mitochondria-Mediated Pathway and the Death Receptor-Induced Pathway Play an Important Role in 9F-Induced Apoptosis

Given the induction of cell death by 9F, as detected using flow cytometry and high-content screening (HCS), we next sought to explore the pathways involved in 9F-induced apoptosis in CRC cells. We observed that genes related to apoptosis were roughly activated in 9F-treated HCT116 cells (Figure 4A), thus confirming the effect of 9F on apoptosis. Similarly, the gene sets regulating cell death induced by p53, the TNF-related apoptosis-inducing ligand (TRAIL) pathway and death receptors were enriched following 9F treatment (Figure 4A), thus providing evidence that the death receptor-mediated pathway contributes to 9F-induced apoptosis. As shown in Figure 4B, 9F treatment effectively promoted the expression of FAS, TNFRSF10A, and TNFRSF10B. Additionally, PIDD1 was increased and BCL2 was reduced in HCT116 cells exposed to 9F for 24 h. To characterize the role of p53 in the death receptor pathway regulated by 9F in CRC cells, TP53 and other genes involved in the death receptor pathway were uploaded into the protein-protein interaction (PPI) online tool Search Tool for the Retrieval of Interacting Genes (STRING) (Figure 4C). Moreover, treatment with 9F increased the expression of Bak, cleaved caspase 3, cleaved caspase 9, death receptor 5, and cleaved caspase 8 in a dose-dependent manner (Figure 4D). These results indicated that 9F may induce apoptosis through intrinsic and extrinsic pathways.

### 2.5. 9F Inhibits Migration and Invasion of CRC Cells

Metastasis is a poor prognostic sign for patients with CRC, and is a positive predictor for CRC-related mortality. Migration and invasion are two key steps that occur during metastasis. We thus explored the biological efficacy of 9F against the motility of CRC cells using modified Boyden chamber assays. As shown in Figure 5A, treatment with 9F decreased the migration of CRC cell lines (CT-26 and HCT116) in a concentration-dependent manner. The quantification of migrated cells confirmed that CRC cells were sensitive to 9F in migration assays. During invasion assays, CRC cells had similar responses to 9F (Figure 5B). The EMT phenotype facilitates the migration and invasion of CRC cells. Thus, we investigated whether 9F might be able to regulate the expression of EMT markers. The downregulation of β-catenin, Snail 1, Vimentin, and N-cadherin, as well as the upregulation of E-cadherin, demonstrated that 9F affected EMT in CRC cells (Figure 5C). To identify the function of p53 in EMT in 9F-treated CRC cells, the STRING database was applied to validate the interactions between p53, β-catenin, Snail 1, Vimentin, N-cadherin, and E-cadherin (Figure 5D). Taken together, these results demonstrated that 9F inhibited EMT, resulting in the suppression of migration and invasion in CRC cells.

### 2.6. 9F Suppresses Proliferation, Migration and Tube Formation of Endothelial Cells

Angiogenesis is essential for tumor growth and metastasis. Therefore, angiogenesis is considered as a therapeutic target for the treatment of cancer. Based on the RNA-seq data, we observed that genes that suppress tumor angiogenesis, including SERPINE1, THBS1, SERPINB5, CD82 and ANGPTL4, were upregulated. By contrast, SLIT2, EFNB2, FGF9, and HEY1, which promote tumor angiogenesis, were downregulated in 9F-treated HCT116 cells (Figure 6A). As shown in Figure 6B, interactions between p53 and these genes were observed. Besides, we found that treatment with 9F upregulated the expression of p53 and p21 in human microvascular endothelial cells (HMEC-1; Figure 6C). These results implied that 9F might affect angiogenesis through p53 signaling. To validate the sensitivity of endothelial cell lines (HUVEC and HMEC-1) to 9F, we first investigated the effects of 9F on the proliferation of HUVEC and HMEC-1 cells. As expected, the cell viability of endothelial cells decreased when cells were exposed to 9F (Figure 6D), which also suggested the toxic effects of 9F on endothelial cells to a certain degree. Moreover, we performed a wound-healing migration assay to examine the migration properties of endothelial cells. We observed a decrease in endothelial migration depending on the concentration of 9F (Figure 6E). Finally, in the tube formation assay, 9F caused a decrease in the formation of tubular structures (Figure 6F). These observations all indicated that 9F inhibited angiogenesis.

### 2.7. 9F Suppresses Tumor Growth and Lung Metastasis of CRC In Vivo

To determine whether 9F suppresses tumor growth in vivo, a xenograft model was established. Mice were treated with physiological saline (control), 9F (0.2 mg/kg and 0.3 mg/kg) and amonafide (5 mg/kg) by intravenous injection. As displayed in Figure 7A, tumor growth in treated groups was retarded. Meanwhile, 9F at a high dose (0.3 mg/kg) yielded better results than amonafide (5.0 mg/kg). The index of body weight of mice demonstrated that there were no dose-associated toxicities (Figure 7B), which was further confirmed by morphological examination of the livers and lungs of the mice from the control and 9F (0.3 mg/kg)-treated group (Figure 7C). Subsequently, we assessed the impact of 9F on the lung metastasis of CRC. As shown in Figure 7D,E, treatment with 9F and amonafide obviously reduced the number of metastasis nodules. Consistent with the tumor growth curve, the high dose of 9Fshowed better treatment effects, as compared to amonafide, in the lung metastasis model. Taken together, these results demonstrated that 9Fwas capable of inhibiting tumor growth and metastasis in vivo.

## 3. Discussion

Previous studies have considered the evaluation of several naphthalimides in clinical trials. However, adverse side effects (e.g., dose-limiting myelosuppression) were observed, which limits the clinical application of amonafide and its derivatives [43,44]. In contrast, 9F has demonstrated lower IC_50_ profiling against hepatocellular carcinoma, compared to amonafide. Of importance, 9F has a lower number of adverse side effects and is safer than amonafide [41]. The biological effects and underlying mechanisms of 9F against CRC have not been reported. In this study, we functionally characterized 9F as more effective than amonafide against CRC in vitro and in vivo. Based on RNA-seq, we revealed that the activation of the p53 signaling was responsible for the efficacy of 9F against CRC.

We found that 9F increased the expression of p53 and the localization of p53 in nuclei (Figure 2). Accumulation of p53, as a nuclear transcriptional factor, triggers transcriptional transactivation of its downstream genes such as p21 and GADD45, resulting in cell cycle arrest [45]. Cell cycle progression is dysregulated in numerous malignant diseases. Therefore, modulation of the cell cycle has been considered as a potential therapeutic strategy for cancer treatment [46]. We have shown that 9F is capable of decreasing the protein expression of Wee1, Cdc25C, Cyclin B, and cyclin E, which are related to theG2/M cell cycle transition (Figure 3). In response to stress, Gadd45A is a key player in G2/M cell cycle arrest. The induction of Gadd45A expression confirmed that G2/M cell cycle arrest was induced in 9F-treated cells. Activation of oncogenes and dysregulation of the cell cycle lead to stalled DNA replication forks [47]. Many factors participate in the process of DNA replication. Among cell cycle regulatory proteins, p21 plays a direct role in inhibiting DNA replication [48]. In addition, E2F and PCNA integrate the cell cycle with DNA replication [49,50]. In eukaryotic cells, the MCM complex (known as DNA replication licensing) controls the once-per-cell cycle DNA replication process [51,52]. Therefore, targeting DNA replication stress is one strategy for cancer treatment. Recent advances have indicated that topoisomerase inhibitors could induce replicative stress by controlling DNA supercoiling and entanglement [53]. Some topoisomerase inhibitors such as camptothecin (CPT), which targets Topo I, and doxorubicin and mitoxantrone, which target Topo II, have been used for cancer therapy in clinical trials [54]. Recently, a series of naphthalimide derivatives have been assessed as topoisomerase inhibitors [8,55]. In line with these findings, our results implied that 9F could be considered as a potential candidate for CRC therapy.

Apoptosis is genetically regulated by the mitochondria-mediated intrinsic pathway and the death receptor-induced extrinsic pathway to control physiological growth and tissue homeostasis [56]. Current cancer therapies primarily trigger apoptosis in cancer cells to exert their anticancer effects. Death receptor genes are likely enhanced through p53-dependent and -independent mechanisms in response to antitumor agents [57,58]. Studies have shown that DNA-damaging agents can increase the expression of DR4 [59], which is upregulated by p53 through an intronic p53 binding site [60]. When cells are exposed to genotoxic stress, DR5 was possibly regulated in a p53-dependent manner [58]. Naphthalimide–polyamine conjugates have been previously shown to induce cell death, partially through the death receptor-mediated pathway [61]. In the present study, we observed that the expression of PIDD, FAS, DR4, and DR5 were increased at mRNA level, while Bcl-2 was reduced in 9F-treated colorectal cells (Figure 4). Meanwhile, the treatment of HCT116 cells induced the expression of Bak, cleaved caspase 3, cleaved caspase 9, DR 5, and cleaved caspase 8. These mechanisms may contribute to 9F-induced apoptosis and suggest that 9F-induced cell death occurs through both intrinsic and extrinsic pathways. In addition, we will investigate the detailed role of death receptors, including DR4 and DR5, in cells treated by 9F (and its derivatives) in our future work.

Cells isolated from p53−/− mice harbor increased the expression of Twist, Snai1, Snai2, Vimentin, Zeb1, and Zeb2, and decreased the expression of E-cadherin [38]. These results indicated that the EMT phenotype was reversed in the absence of p53. Valentini et al. found that the accumulation of β-catenin was observed in CRCs with high frequency mutations of p53 [62]. Sadot et al. also observed that the activation of p53 resulted in the downregulation of β-catenin [63], a marker of EMT, which promotes the migration and invasion of CRC [64]. Our results indicated that 9F inhibited the migration and invasion of CRCs by altering the expression of EMT markers (Figure 5). Naphthalimide–polyamine conjugates have been evaluated in their ability to prevent the migration and invasion of cancer cells and, subsequently, to suppress metastasis in vivo [40,42]. In addition, treatment with 9F induced cell cycle arrest and proliferation inhibition affects the cell viability, resulting in the partial inhibition of EMT. Thus, the effects of topoisomerase inhibitors, such as amonafide and its derivatives, on EMT should be investigated under lower concentrations (which will not induce cell cycle arrest or DNA damage) in future work. Taken together, inhibition of EMT regulated by p53 may be responsible for the improved therapeutic response to 9F through the suppression of metastasis of CRC in vivo (Figure 7).

Tumor angiogenesis is a prognostic factor used to characterize CRC patient outcome. Emerging studies have revealed that the microvascular density of CRC is related to tumor growth and metastasis [65,66]. Hence, targeting angiogenesis has been considered as an attractive strategy for CRC treatment. To block tumor angiogenesis, angiogenesis inhibitors have been developed intensively. These drugs range from endogenous angiogenesis inhibitors to monoclonal antibodies as well as small molecules. Some of them have been successfully integrated into current treatment regimens and have displayed efficacy in CRC treatment [67]. Many proteins such as thrombospondins, vasohibin, and chondromodulin, have been identified as endogenous angiogenesis inhibitors [68,69,70]. Our RNA-seq results indicated that 9F treatment induced the transcript levels of thrombospondin 1, SERPINB5, CD82, and SERPINE1 (Figure 6), which have been reported to display anti-angiogenic activities upon p53 activation [71]. In this study, we found that endothelial cells are sensitive to treatment with 9F, and that expression of p53 and p21 is upregulated in 9F-treated endothelial cells (Figure 6). The inhibitory effects of naphthalimide-polyamine conjugates on angiogenesis have rarely been reported. Analyzing the activity of other naphthalimide-polyamine conjugates against angiogenesis will be necessary to determine whether angiogenesis is involved in the suppressive effects of naphthalimide-polyamine conjugates on CRC.

## 4. Materials and Methods

### 4.1. Cell Lines and Reagents

Colorectal cancer cells lines HCT8, CT-26, Human umbilical vein endothelia cell (HUVEC) and human microvessel endothelial cell-1 (HMEC-1) were obtained from the American Type Culture Collection (ATCC, Manassas, VA, USA). HCT116 p53−/− and HCT116 p53+/+ were kind gifts from Prof. Xiaotao Li, East China Normal University. Colorectal cancer cells were cultured in RPMI-1640 medium (Thermo Fisher Scientific, Waltham, MA, USA) supplemented with 10% FBS (PAN-Seratech, Aidenbach, Germany), streptomycin (100 units/mL), and penicillin (100 units/mL) at 37 °C in 5% CO_2_. 9F was synthesized in our laboratory.

### 4.2. MTT Assay

Cell viability was measured using MTT assay [72]. In brief, 4 × 10^3^ cells were seeded into 96-well plates and exposed to different concentrations of 9F or amonafide for the indicated time. After treatment, 50 μL MTT (1 mg/mL) was added followed by incubation at 37 °C and 100 μL DMSO was added to dissolve the crystal products. Absorbance was measured using a microtiter plate reader, and all experiments were repeated at least three times. Antibodies were purchased from Cell Signaling Technology Inc (Danvers, MA, USA).

### 4.3. 2D Colony Formation Assay

The 2D colony formation assay was carried out for measurement of clonogenic cell survival. Briefly, 500 cells in 100 μL medium were seeded into the wells of 96-well plates. Cells were cultured for 3 days and then treated with 9F at various concentrations for another 7 days. Cells were washed with PBS, fixed with 4% paraformaldehyde for 20 min, and stained with crystal violet. Images were taken and colonies were counted. All experiments were repeated three times.

### 4.4. Apoptosis Detection

AO/EB staining and Annexin V-FITC/PI staining were used to examine the apoptosis of colorectal cancer cells as previous description. In brief, after cells were treated with 9F for 48 h, cells were stained with AO/EB or Annexin V-FITC/PI, and the fluorescence of AO/EB and Annexin V-FITC/PI were determined by high content screening (HCS) and flow cytometry, respectively (BD Biosciences, San Jose, CA, USA).

### 4.5. Cell Cycle Assay

HCT116 cells and CT-26 cells were treated by 9F for 48 h, then harvested and washed with PBS. Cells were fixed with 70% cold ethanol overnight at 4 °C. Cells were collected and re-suspended with PBS containing RNase. After incubation at 37 °C for 30 min, cells were stained with PI and analyzed using flow cytometry.

### 4.6. Immunofluorescence Assay

HCT116 cells were treated with 9F for 48 h, and fixed by 4% formaldehyde. Fixed cells were treated with 0.2% Triton-X 100 and blocked using 0.5% FBS. After incubation with the anti-p53 antibody overnight at 4 °C, cells were treated with the secondary antibody. Then, Hoechst33342 was added to stain the nucleus. Images were taken by confocal microscopy (Nikon, Tokyo, Japan).

### 4.7. Western Bloting Assay

Cells were treated with 9F at various concentrations for 48 h and harvested. Collected cells were washed with cold PBS and lysed using RIPA buffer (Beyotime, Shanghai, China). A BCA assay kit was used to detect the concentration of proteins [73]. The samples were subjected to SDS-loading buffer at 100 °C for 10 min and exposed to SDS-PAGE. Proteins were transferred onto PVDF membranes which were subsequently blocked by 5% dried skimmed milk. The expression of protein was detected using the ECL plus reagents (Beyotime).

### 4.8. Boyden Chamber Migration and Invasion Assay

A boyden chamber system was used to assess colorectal cancer cell migration and invasion. After starvation for 8 h, cells were seeded to the chamber with 8-μm pore. Images were taken after 9F treatment for 24 h, and migrated cells were recorded. For the invasion assay, the chamber was pre-coated with Matrigel.

### 4.9. Angiogenesis Assay

A wound healing migration assay and tube formation assay were performed to evaluate the effect of 9F on angiogenesis [74]. Endothelial cells were cultured in 6-well plates pre-coated with 1% gelatin. Cells were scratched with a pipette tip followed by the addition of 9F at various concentrations. Images of the wound area were taken after 9F treatment for 8 h, and the number of migrated cells was recorded. As for the tube formation assay, 96-well plates were pre-coated with Matrigel, and 1 × 10^4^ endothelial cells per well were seeded. After incubation for 8 h, tube structures were photographed and counted.

### 4.10. Mice, Tumor Model and Treatment

All animal experiments performed in this article were conducted in compliance with the Guide for the Care and Use of Laboratory Animals published by Henan University (protocol code: HUSOM-2019-162). Six-week-old Balb/c mice were injected subcutaneously with CT-26 cells (5 × 10^5^ cells per mouse). After 3 days, mice (6 mice/each group) were treated with physiologic saline (control), 9F (0.2 mg/kg and 0.3 mg/kg) and amonafide (5 mg/kg) via the tail vein for 18 consecutive days. Body weight and tumor dimensions were measured performed daily. The tumor volume was considered the volume equal to 0.52 × AB^2^, where A is the long axis and B is the short axis. CT-26 cells (1 × 10^5^ cells per mouse) were injected via the tail vein to establish lung metastasis model. To ensure the establishment of lung metastasis, the tumor was incubated for 3 days. Next, mice (5 mice/each group) were treated with physiologic saline (control), 9F (0.2 mg/kg and 0.3 mg/kg) and amonafide (5 mg/kg) via the tail vein every day. After treatment for 21 days, mice were anesthetized and euthanized. Lungs were removed, and the nodules were counted.

### 4.11. Hematoxylin and Eosin Staining

Tissues from the mice were fixed by 4% paraformaldehyde and sectioned at 5 μm, haematoxylin and eosin staining was performed according to a standard protocol [41].

### 4.12. RNA-Sequencing and Bioinformatics Analysis

According to the Wuhan Huada Sequencing Company’s (Wuhan, China) instructions, the cDNA library construction, library purification and transcriptome sequencing were performed. Gene Ontology (GO) terms, Kyoto Encyclopedia of Genes and Genomes (KEGG), REACTOME, and gene set enrichment analysis (GSEA) were utilized. Search Tool for the Retrieval of Interacting Genes (STRING) was used to predict protein-protein interactions (PPI).

### 4.13. Statistical Analysis

In this study, data were expressed as mean ± SD from at least three separate experiments. Unless otherwise noted, the differences between groups were analyzed using Student‘s *t*-test. *p* < 0.05 was set to be considered statistically significant, * *p* < 0.05, ** *p* < 0.01, and *** *p* < 0.001.

## 5. Conclusions

Our findings demonstrated that 9F induces apoptosis and cell cycle arrestin CRC cells, and suppresses the migration and invasion of CRC cells (as well as angiogenesis) in vitro, leading to the inhibition of tumor growth and metastasis in vivo. According to bioinformatic analyses based on RNA-seq data, we found that 9F regulates the expression of key regulators of the cell cycle, apoptosis, the epithelial-to-mesenchymal transition (EMT), and angiogenesis through p53 signaling. In addition, 9F is more effective than amonafide against CRC in vitro and in vivo.

In summary, we have demonstrated the key role of the p53-mediated pathway in examining the therapeutic response of CRC cells to 9F in vitro and in vivo. Our findings warrant the further development of 9F as a potential therapeutic agent for patients with CRC.

## Figures and Tables

**Figure 1 cancers-12-00528-f001:**
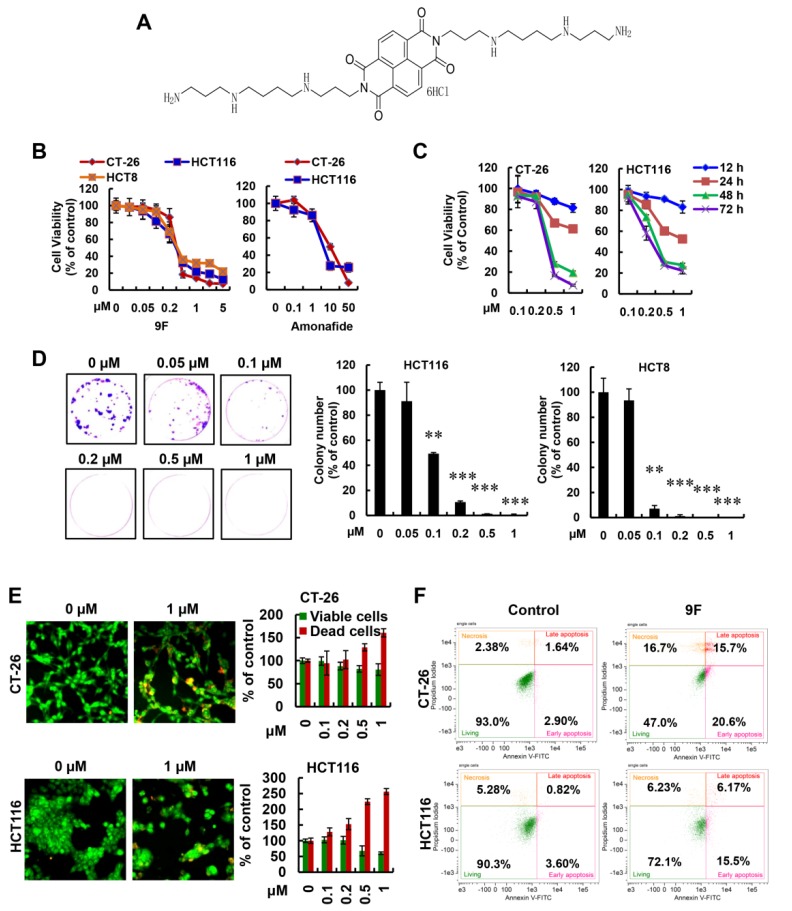
9F inhibited cell growth and induced cell death of colorectal cancer (CRC) cells. (**A**) The chemical structure of 9F. (**B**) Cell viability was measured in CRC cells treated with various concentrations of 9F and amonafide for 48 h. (**C**) Cell viability was measured in CRC cells treated with various concentrations of 9F for 24 h, 48 h and 72 h. (**D**) Representative images (HCT116 cells) and statistics results of colony formation assay in CRC cells. CRC cells were treated with 9F at different concentrations for 7 days. ** *p* < 0.01, and *** *p* < 0.001. (**E**) AO/EB staining was used to detect cell death of CRC cells exposed to 9F for 48 h (20×). (**F**) CT-26 cells and HCT116 cells were treated with 9F (1 μM) for 48 h. Cell death was detected using flow cytometry.

**Figure 2 cancers-12-00528-f002:**
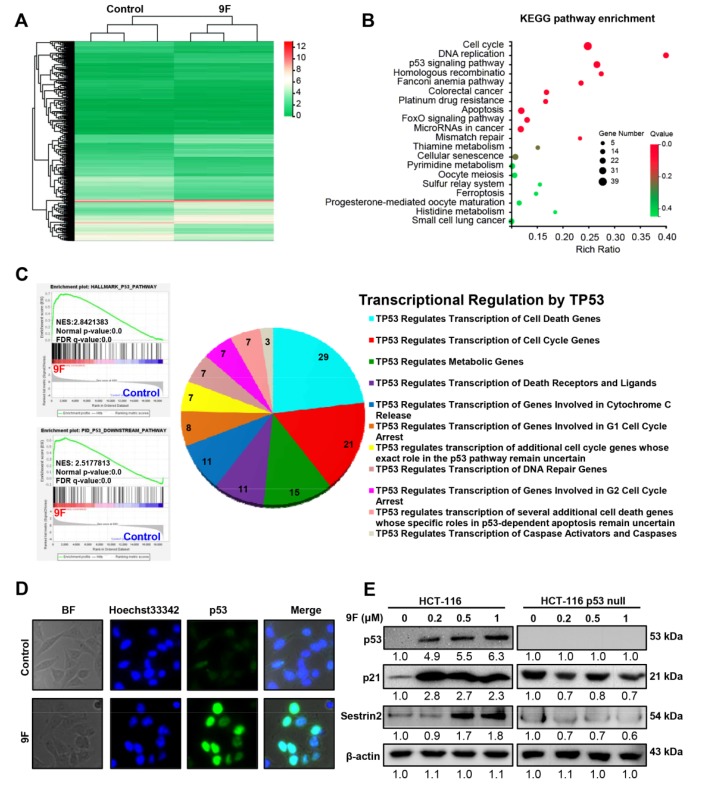
p53 signaling was altered in 9F-treated CRC cells. (**A**) Cluster heatmap of differentially expressed genes in 9F-treated HCT116 cells. Cells were treated by 9F (1 μM) for 24 h. (**B**) KEGG pathway enrichment analysis of differentially expressed genes (DEGs) were performed. The top20 pathways were shown. (**C**) Gene set enrichment analysis (GSEA) plots correlating with p53 pathway and p53 downstream pathway (left), and Reactome analyses of transcriptional regulation by p53 (right). (**D**) Immunofluorescence staining showing localization of p53 in nuclear in HCT116 cells treated by 9F for 48 h (63×). (**E**) Western blot analyses showing p53, p21 and Sestrin 2 protein levels in p53-wildtype HCT116 cells and p53-null HCT116 cells. Cells were treated by 9F for 48 h. β-actin was used as loading control.

**Figure 3 cancers-12-00528-f003:**
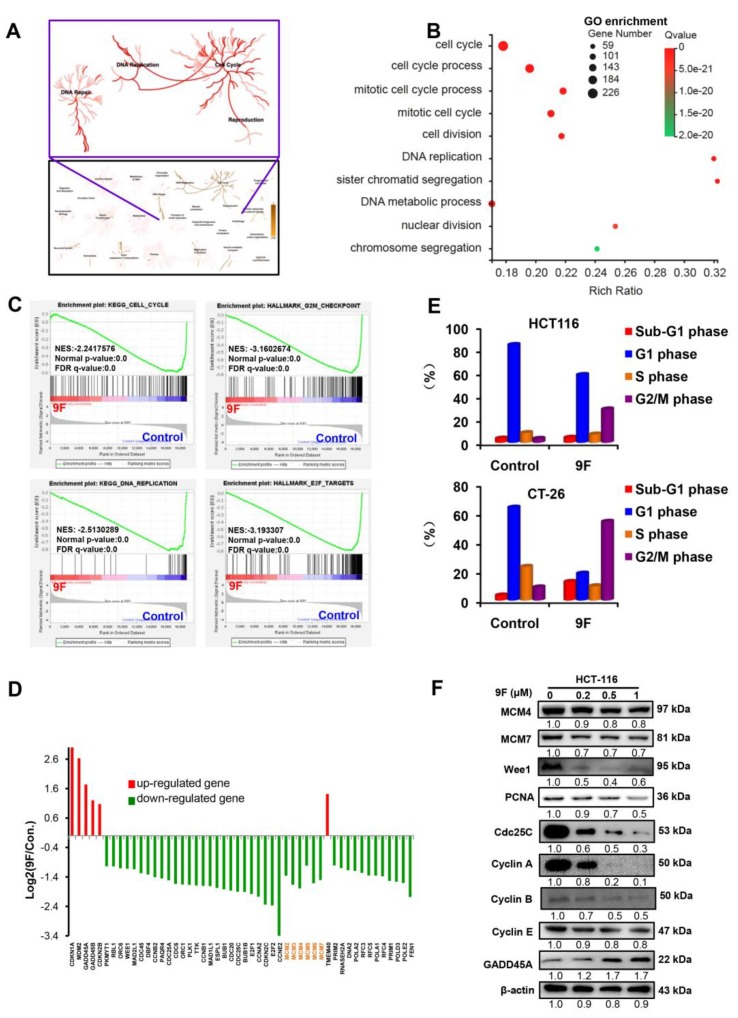
9F induced G2/M cell cycle arrest in CRC cells. (**A**) Reactome pathway database was used to analyse DEGs in 9F-treated HCT116 cells. (**B**) GO enrichment analyses of DEGs in 9F-treated HCT116 cells. (**C**) GSEA plots correlating with cell cycle, DNA replication, G2/M checkpoint and E2F targets. (**D**) The transcript levels of genes related with cell cycle and DNA replication from RNA-seq data. (**E**) CT-26 cells and HCT116 cells were exposed to 9F for 48 h. The cells were stained using PI, and the cell cycle progression was analyzed by flow cytometry. (**F**) Western blot analyses of cell cycle and DNA replication regulatory proteins following 9F treatment for 48 h.

**Figure 4 cancers-12-00528-f004:**
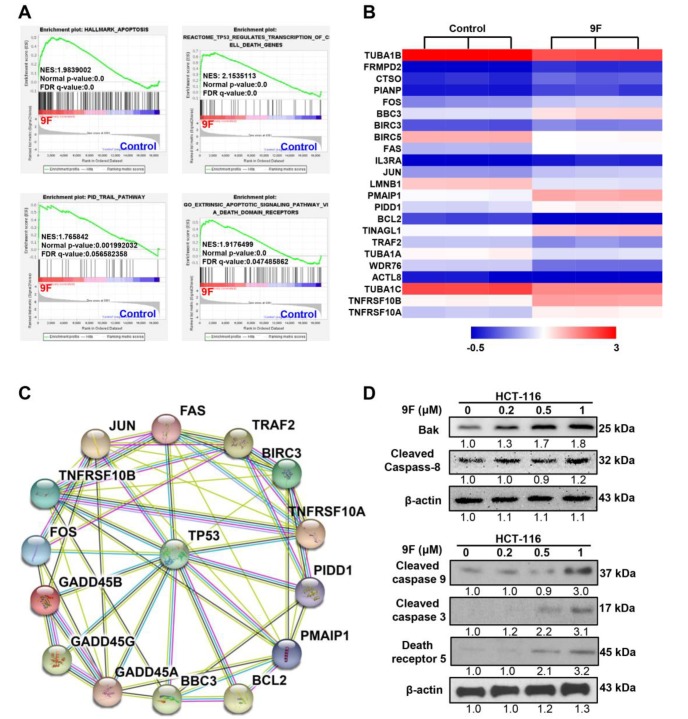
Activation of intrinsic and extrinsic apoptosis pathways accounted for 9F-induced cell death. (**A**) GSEA plots correlating with apoptosis, p53 regulates transcription of cell death genes, TNF-related apoptosis-inducing ligand (TRAIL) pathway and extrinsic apoptotic signaling pathway via death domain receptors. (**B**) Heatmap of differentially expressed genes involved in apoptosis. (**C**) Protein-protein interaction (PPI) network based on Search Tool for the Retrieval of Interacting Genes (STRING) showing the detailed interactions of p53 and cell death-related proteins. (**D**) Western blot analyses of Bak, cleaved caspase 8, cleaved caspase 3, cleaved caspase 9 and death receptor 5 following 9F treatment for 48 h.

**Figure 5 cancers-12-00528-f005:**
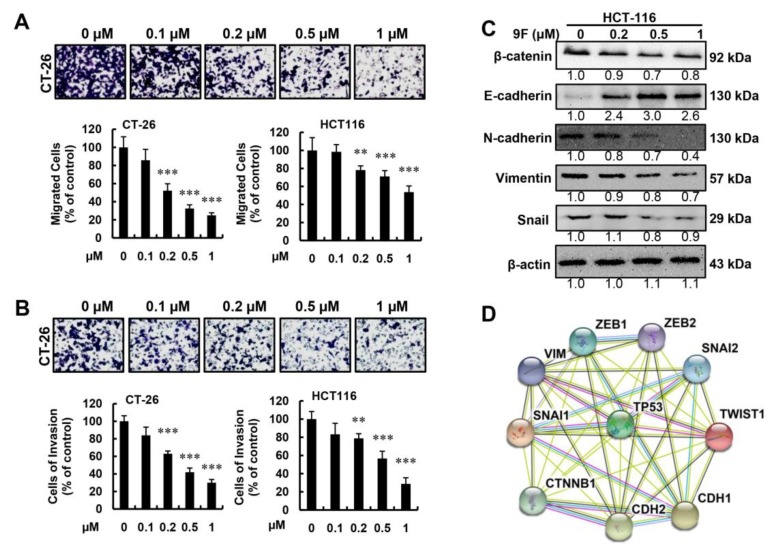
The abilities of migration and invasion were lost in CRC cells exposed to 9F. (**A**) Cell migration of CT-26 cells and HCT116 cells was analyzed by the Boyden chamber assay (20×). ** *p* < 0.01, and *** *p* < 0.001. (**B**) Cell invasion of CT-26 cells and HCT116 cells was analyzed by the Boyden chamber assay (20×). ** *p* < 0.01, and *** *p* < 0.001. (**C**) Western blot analyses of epithelial-to-mesenchymal transition (EMT) markers in HCT116 cells following 9F treatment for 48 h. (**D**) PPI network based on STRING showing the detailed interactions of p53 and EMT markers.

**Figure 6 cancers-12-00528-f006:**
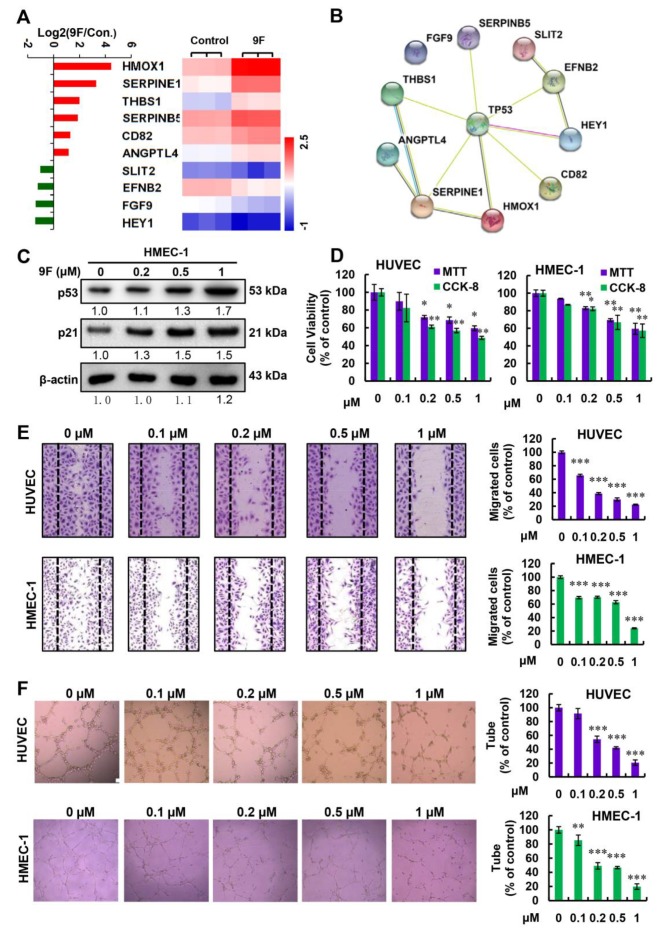
9F suppressed proliferation, migration and tube formation of endothelial cells. (**A**) The transcript levels of genes related to angiogenesis from RNA-seq data. (**B**) PPI network based on STRING showing the detailed interactions of p53 and the angiogenesis-related genes regulated by 9F treatment. (**C**) Western blot analyses of p53 and p21 protein in 9F-treated HMEC-1 cells. Cells were treated by 9Ffor 48 h. (**D**) Cell viability was measured in human endothelial cells treated with various concentrations of 9F for 48 h. * *p* < 0.05, ** *p* < 0.01. (**E**) Cell migration of endothelial cells was analyzed by the wound healing assay (10×). *** *p* < 0.001. (**F**) Tube formation by endothelial cells was assayed after 9F treatment for 8 h (10×). ** *p* < 0.01, and *** *p* < 0.001.

**Figure 7 cancers-12-00528-f007:**
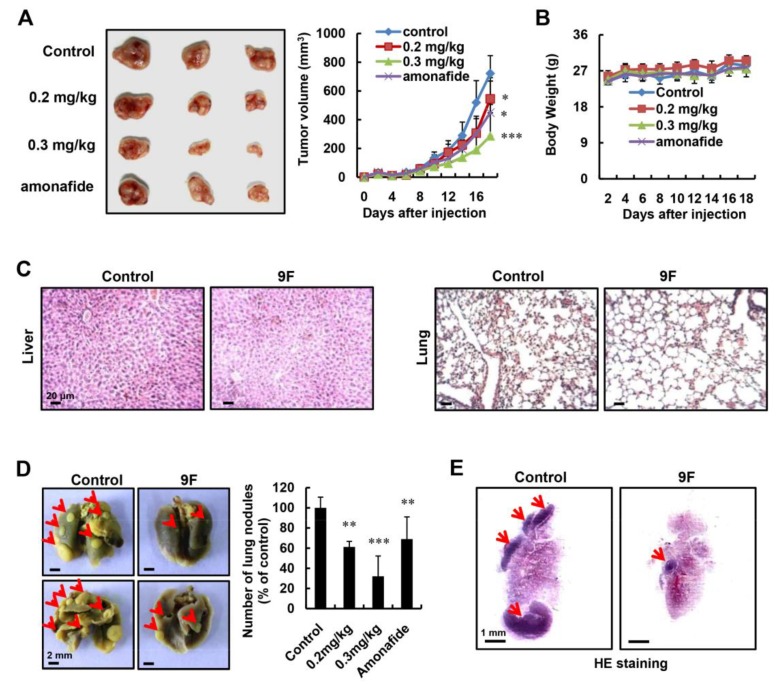
Treatment with 9F inhibited tumor growth and metastasis of CRC in vivo. (**A**) Tumor volumes of mice treated by 9F and amonafide were evaluated on the indicated days. Amonafide was used as positive control. * *p* < 0.05, *** *p* < 0.001. (**B**) Body weights of mice were measured. (**C**) H&E staining of liver and lung from mice bearing CRC cells after treatment with or without 9F (0.3 mg/kg). (**D**) The representative images of lung metastasis nodules from mice bearing CRC cells after treatment with or without 9F (0.3 mg/kg). The number of metastasis nodules was recorded. Amonafide was used as positive control. ** *p* < 0.01, and *** *p* < 0.001. (**E**) The representative images of H&E staining of lung metastasis from mice treated with or without 9F (0.3 mg/kg).

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
