# Peer review of "The Role of p53-Mediated Signaling in the Therapeutic Response of Colorectal Cancer to 9F, a Spermine-Modified Naphthalene Diimide Derivative"

_cancers, 2020, doi:10.3390/cancers12030528_

Round 1
Reviewer 1 Report
In the current manuscript, the authors investigated the effect of 9F, a spermine-modified naphthalene diimide derivative, on colorectal cancer. They found that 9F induces apoptosis, cell cycle arrest and suppresses migration, invasion and angiogenesis in vitro. They also confirmed their findings in mouse model. Most important, RNA-seq results indicate 9F exerts its antitumor effect via p53 induction. Overall, it’s an interesting paper. Some minor issues should be rechecked. Specific points are included as follows:
In Figure 1B and 1C, 9F suppresses human and mouse colon cancer cells viability in a time- and does-dependent manner. Have the authors examined the effect of 9F on normal colonic epithelial cells? In Figure 1F. 9F induces high population Annexin V-/PI+ in CT26 cells, which means necroptosis was induced. The authors should pay attention on 9F-induced necroptosis in future. In Figure 2E. 9F promotes p53 induction in HCT116 cells. In addition, p53 nuclear translocation was increased upon 9F treatment. Thus, the authors should investigate the effect of 9F on the mechanism of p53 translocation. In Figurer 4. The authors investigated the effect of 9F on apoptosis. The increased of cleaved caspase 8 was not significant (Figure 4D). In addition, caspase 9 activation, an intrinsic apoptotic pathway indicator, should be examined. In Figure 5A and 5B. The authors should point out what cell line the picture presents? Scal bar should be provided in Figure 7C and 7E. References are missing in some place, please add references correctly. The information of primary antibodies should be provided.
Author Response
Dear Reviewer,
Thank you very much for your positive comments and insightful suggestions. It’s our great pleasure to submit our revised manuscript entitled “The role of p53-mediated signaling in the therapeutic response of colorectal cancer to 9F, a spermine-modified naphthalene diimide derivative” (Manuscript ID: cancers-695698) by Lei Gao et al. to Cancers. In this revision, we have tried our best to make changes requested by you. In addition, we have also modified some minor errors, such as grammar and so on, in our revised manuscript. A document answering every question was summarized and enclosed.Please see the attachment. With your help, we hope that the quality of this manuscript has been substantially improved for publication in Cancers.
Thank you very much for all your help again.
Best regards
Sincerely yours
Fujun Dai

Reviewer 2 Report
In the present study, authors reported that 9F, which is a naphthalene diimide derivative with polyamine moieties, which inhibits the growth and lung metastasis of CRC, in vitro and in vivo. By using RNA-seq, authors conclude that 9F induce cell cycle arrest, apoptosis and oppose EMT as well as angiogenesis by activating p53-mediated signaling.
Major points
Although data are interesting, in general data are over-interpreted and conclusions are overstated. Apoptosis data by FACs analysis (Fig 1F) should be correctly interpreted and conclusions adjusted. The authors shown data of mixed apoptotic and necrotic cells in the 9F-FACs analysis. Therefore, apoptosis in the 9F assays could not clearly supported, in my opinion should be simply considered as cell death. In addition, FACs panels were analyzed in the wrong way. See example in:
https://www.raybiotech.com/raybio-annexin-v-apoptosis-kit-raybright-v450/
https://www.bdbiosciences.com/documents/BD_FACSVerse_Apoptosis_Detection_AppNote.pdf
3. Apoptosis is an early event it occurs between 8-12h post drug insult; late apoptosis normally occurs during mitosis around 18-24h, therefore apoptosis assays should be measured before cell duplication (HCT116= 18h; CT-26= 15-20h), ideally apoptosis assays should be before 24h to avoid any misinterpretations. Apoptosis positive controls are required. Moreover, AnexinV is an early marker of apoptosis; therefore, using this marker at 48h post stimulus is an experimental flaw. Indeed, Caspase-3 processing and PARP cutting should be used, as late apoptosis markers.
4. Interpretation of the expression profiling data by RNA seq analysis is difficult, since amounts of 9F used were not indicated in figure 2. Obviously, the expression data are produced by surviving cells, and data should be discussed in this sense.
5. It is speculated from expression profiling data that G2/M arrest occurs in the treated cells, however, ciclin B are down regulated by using concentrations lower than 05. µM of 9F, this data is inconsistent with a G2/M arrest; the data sustained G2/M arrest only for the concentration of 1 µM. However, it is not indicated in the manuscript the concentration of 9F used for RNA seq analysis.
6.a) The conclusion that exposition to 9F promotes extrinsic apoptosis is not sustained by the data, there is no evidence in the manuscript that TRAIL receptors (1 or2) ligands are produced by CT-26 or HCT116 cells. Authors cited only their own previously published data to supports this conclusion.
b) processed caspase 8 levels are shown in figure 4D, as proof of caspase-8 activation, however, the activity of this caspase is difficult to assess, even by measuring its activity by biochemical methods. This data should be verified by using substrate specific activity of caspase-8, being 9F a topoisomerase inhibitor, the most important cause of apoptosis is DNA damage and therefore, cell death should be triggered by intrinsic apoptosis.
7. This reviewer ignore whether data of EMT and migration data is relevant, since the 9F insulted cells are probably arrested and intending to repair the DNA damage produced by the topoisomerase inhibition.
8. Similarly, is difficult to assess the relevance of the experiments of HMEC-1 cells treated with 9F, since they are normal and not cancer cells. These Data suggested toxic effectsof 9F, for normal cells
9. Expression of TNFRs and Death receptor should be confirmed by western blot and its functionality should be verified, since many cancer cells present cell death resistance by extrinsic immune ligands.
Author Response
Dear Reviewer,
Thank you very much for your positive comments and insightful suggestions. It’s our great pleasure to submit our revised manuscript entitled “The role of p53-mediated signaling in the therapeutic response of colorectal cancer to 9F, a spermine-modified naphthalene diimide derivative” (Manuscript ID: cancers-695698) by Lei Gao et al. to Cancers. In this revision, we have tried our best to make changes requested by you. In addition, we have also modified some minor errors, such as grammar and so on, in our revised manuscript. A document answering every question was summarized and enclosed. Please see the attachment. With your help, we hope that the quality of this manuscript has been substantially improved for publication in Cancers.
Thank you very much for all your help again.
Best regards
Sincerely yours
Fujun Dai

Round 2
Reviewer 2 Report
The manuscript was improved